# Long-term health outcomes of people with reduced kidney function in the UK: A modelling study using population health data

Iryna Schlackow[1☯], Claire Simons[1☯], Jason Oke[2], Benjamin Feakins[2], Christopher A. O'Callaghan[3], F. D Richard Hobbs[2], Daniel Lasserson[4], Richard J. Stevens[2], Rafael Perera[2], Borislava Mihaylova[1,5]*

1 Nuffield Department of Population Health, University of Oxford, Oxford, United Kingdom, 2 Nuffield Department of Primary Care Health Sciences, University of Oxford, Oxford, United Kingdom, 3 Nuffield Department of Medicine, University of Oxford, Oxford, United Kingdom, 4 Warwick Medical School, Population Evidence and Technologies, University of Warwick, Warwick, United Kingdom, 5 Institute of Population Health Sciences, Barts and The London School of Medicine and Dentistry, Queen Mary University of London, London, United Kingdom

☯ These authors contributed equally to this work.
* boby.mihaylova@dph.ox.ac.uk

**Data Availability Statement:** The data were obtained from the Clinical Practice Research Datalink (CPRD). CPRD data governance does not

## Abstract

### Background

People with reduced kidney function have increased cardiovascular disease (CVD) risk. We present a policy model that simulates individuals' long-term health outcomes and costs to inform strategies to reduce risks of kidney and CVDs in this population.

### Methods and findings

We used a United Kingdom primary healthcare database, the Clinical Practice Research Datalink (CPRD), linked with secondary healthcare and mortality data, to derive an open 2005–2013 cohort of adults (≥18 years of age) with reduced kidney function (≥2 measures of estimated glomerular filtration rate [eGFR] <90 mL/min/1.73 m$^2$ ≥90 days apart). Data on individuals' sociodemographic and clinical characteristics at entry and outcomes (first occurrences of stroke, myocardial infarction (MI), and hospitalisation for heart failure; annual kidney disease stages; and cardiovascular and nonvascular deaths) during follow-up were extracted. The cohort was used to estimate risk equations for outcomes and develop a chronic kidney disease–cardiovascular disease (CKD–CVD) health outcomes model, a Markov state transition model simulating individuals' long-term outcomes, healthcare costs, and quality of life based on their characteristics at entry. Model-simulated cumulative risks of outcomes were compared with respective observed risks using a split-sample approach. To illustrate model value, we assess the benefits of partial (i.e., at 2013 levels) and optimal (i.e., fully compliant with clinical guidelines in 2019) use of cardioprotective medications. The cohort included 1.1 million individuals with reduced kidney function (median follow-up 4.9 years, 45% men, 19% with CVD, and 74% with only mildly decreased eGFR of 60–89 mL/min/1.73 m$^2$ at entry). Age, kidney function status, and CVD events were the key

allow us to distribute patient data to other parties. Researchers may apply for data access at http://www.CPRD.com/. Software and codelists availability The R code of the CKD-CVD policy model (License: MIT License) and the codelists to specify disease endpoints and patient characteristics using CPRD data are available at http://www.herc.ox.ac.uk/downloads/supportingmaterial. The R code of the CKD-CVD policy model is also available from https://github.com/ischlackow/CKD-CVD-policy-model. The standalone R code of the CKD progression model is provided in the CKD submodel manuscript[20].

**Funding:** The research was funded by the National Institute for Health Research (NIHR) Programme Grants for Applied Research programme (Reference: RP-PG-1210-12003); with RP as the PI and RS as a co-PI; website https://www.nihr.ac.uk/explore-nihr/funding-programmes/programme-grants-for-applied-research.htm RP, RH and BM acknowledge support by the NIHR Oxford Biomedical Research Centre. https://oxfordbrc.nihr.ac.uk/ RP and RH receive funding from the NIHR Collaboration for Leadership in Health Research and Care (CLARHC; https://www.clahrc-oxford.nihr.ac.uk/) Oxford and the NIHR Oxford Medtech and In-Vitro Diagnostics Co-operative (MIC; https://www.community.healthcare.mic.nihr.ac.uk/). DL receives funding from the NIHR Applied Research Collaboration (ARC) West Midlands and NIHR Oxford MIC, hosted by Oxford Health NHS Foundation Trust. The funders had no role in study design, data collection and analysis, decision to publish, or preparation of the manuscript.

**Competing interests:** I have read the journal's policy and the authors of this manuscript have the following competing interests: RP declares being the PI of NIHR funded Programme of work to evaluate how to monitor CKD in Primary Care which part-funded his salary as well as members of the research team.

**Abbreviations:** AIC, Akaike information criterion; BMI, body mass index; CI, confidence interval; CKD, chronic kidney disease; CKD–CVD, chronic kidney disease–cardiovascular disease; CKD-EPI, Chronic Kidney Disease Epidemiology Collaboration; CPRD, Clinical Practice Research Datalink; CVD, cardiovascular disease; eGFR, estimated glomerular filtration rate; HES, Hospital Episodes Statistics; ICD-10, International Statistical Classification of Diseases and Related Health Problems 10th Revision; ISAC, Independent Scientific Advisory Committee; KDIGO, Kidney Disease Improving Global Outcomes; MI, myocardial infarction; NICE, National Institute for Health and Care Excellence; ONS, Office for

determinants of subsequent morbidity and mortality. The model-simulated cumulative disease risks corresponded well to observed risks in participant categories by eGFR level. Without the use of cardioprotective medications, for 60- to 69-year-old individuals with mildly decreased eGFR (60–89 mL/min/1.73 m$^2$), the model projected a further 22.1 (95% confidence interval [CI] 21.8–22.3) years of life if without previous CVD and 18.6 (18.2–18.9) years if with CVD. Cardioprotective medication use at 2013 levels (29%–44% of indicated individuals without CVD; 64%–76% of those with CVD) was projected to increase their life expectancy by 0.19 (0.14–0.23) and 0.90 (0.50–1.21) years, respectively. At optimal cardioprotective medication use, the projected health gains in these individuals increased by further 0.33 (0.25–0.40) and 0.37 (0.20–0.50) years, respectively. Limitations include risk factor measurements from the UK routine primary care database and limited albuminuria measurements.

## Conclusions

The CKD–CVD policy model is a novel resource for projecting long-term health outcomes and assessing treatment strategies in people with reduced kidney function. The model indicates clear survival benefits with cardioprotective treatments in this population and scope for further benefits if use of these treatments is optimised.

## Author summary

### Why was this study done?

- Chronic kidney disease (CKD) is highly prevalent, and even mildly reduced kidney function increases cardiovascular and kidney disease risks and mortality.

- In people with CKD, reducing cardiovascular risk with widely available effective cardioprotective treatments (i.e., statins, hypertensives, and antiplatelets) is a key target.

- Lifetime policy models are needed to project long-term health outcomes and costs and prioritise treatment strategies.

### What did the researchers do and find?

- We used a large UK population healthcare database to identify a large open cohort (2005–2013) of 1.1 million individuals with reduced kidney function.

- We developed a policy model that projects the decline of kidney function, cardiovascular disease (CVD), mortality, healthcare costs, and quality of life using an individual's characteristics.

- The model achieved good risk discrimination and accurately predicted risks of cardiovascular events in patient categories by kidney function impairment (estimated glomerular filtration rate [eGFR] 60–89; 45–59; 30–44; 15–29; and <15 mL/min/1.73 m$^2$ not on renal replacement therapy [RRT]) and by 10 geographic regions in England.

National Statistics; QALY, quality-adjusted life year;
RRT, renal replacement therapy; TRIPOD,
Transparent Reporting of a multivariable prediction
model for Individual Prognosis Or Diagnosis;
uACR, urinary albumin-to-creatinine ratio.

- To illustrate model use in this population, we assessed survival benefits with partial (0.07–1.50 extra years per person across patient categories) and optimal (0.10–0.61 extra years per person across patient categories) use of cardioprotective treatments.

### What do these findings mean?

- The model can be used to project long-term health outcomes in people with reduced kidney function and assess value of a range of treatment strategies. Further efforts to improve the use of cardioprotective medication are likely to improve life expectancy in this population.

## Introduction

Chronic kidney disease (CKD) affects over 250 million people worldwide [1,2], with prevalence expected to increase with rising levels of obesity and diabetes and ageing populations. This progressive disease is associated with increased risk of cardiovascular disease (CVD) and all-cause mortality [3–5], and most people with CKD die before reaching end-stage renal disease [6]. Excess CVD mortality has been reported even in people with only mildly reduced kidney function and prior to the clinical diagnosis of CKD [7]. Since kidney function typically declines slowly over time, people may spend many years with mild or moderate disease [8] and at increased CVD risk. Timely CVD prevention is a key treatment target, with growing efforts to optimise treatments and develop new therapies across the spectrum of people with kidney impairment. Long-term disease models are, therefore, needed to guide the assessment of the net effects and cost-effectiveness of different management regimens.

The available long-term models for people with reduced kidney function, however, are either derived from a range of literature sources with limited ability to assess variation of effects across people at different disease risks [9–13] or consider only populations with moderate to advanced CKD [5], whereas the vast majority of patients seen in primary care have only mildly reduced kidney function (estimated glomerular filtration rate [eGFR] of 60 to 89 mL/min/1.73 m$^2$ [14]). Routine healthcare data are now available to inform models of the necessary complexity to address key policy questions across the spectrum of patients with reduced kidney function.

We present an internally validated lifetime policy model for people with reduced kidney function (eGFR <90 mL/min/1.73 m$^2$), developed using the detailed individual patient data from a UK population-based primary care database, the Clinical Practice Research Datalink (CPRD), linked with secondary healthcare data (Hospital Episodes Statistics [HES]), mortality registries, and social deprivation data. The model overcomes many of the limitations of previous models and projects decline of kidney function, experience of cardiovascular events and mortality, as well as health-related quality of life and healthcare costs across the range of people with reduced kidney function.

## Methods

This study is reported as per Transparent Reporting of a multivariable prediction model for Individual Prognosis Or Diagnosis (TRIPOD) Checklist for Prediction Model Development and Validation (S1 Checklist) [15].

## Data

The CPRD is a database of routine primary care records in the UK (674 practices; 11.3 million people) [16]. For the present analyses, data from the 388 practices in England with CPRD data linked to (a) UK HES, containing dates of admission and the International Statistical Classification of Diseases and Related Health Problems 10th Revision (ICD-10) codes at discharge for each hospital episode, including up to 99 secondary discharge codes; (b) mortality data from Office for National Statistics (ONS), containing up to 15 ICD-10 codes for contributing causes of death; and (c) Townsend socioeconomic deprivation quintiles were used.

## Study population

We derived an open cohort of adult patients ($\geq$18 years of age) who were registered at "up-to-standard" CPRD practices (i.e., meeting the CPRD standard for continuity of recording and number of recorded deaths) between January 1, 2005 and December 31, 2013 and were deemed to have acceptable patient records (based on continuous registration status, good quality recording of events, and valid age and gender). Patients who were pregnant in the 12 months preceding cohort entry or have had renal transplantation or were on maintenance dialysis at any time prior to cohort entry were excluded. To be included in the study cohort, a patient had to have (at least) 2 eGFR tests <90 mL/min/1.73 m$^2$ at least 90 days apart, in line with the definitions used in Kidney Disease Improving Global Outcomes (KDIGO) classification [14] and the National Institute for Health and Care Excellence (NICE) [17]. All available data after (the latest of) the patient's current registration date, the date the practice became up to standard, and prior to study start date were used to define eligibility. To ensure adequate recording of baseline covariates, eligible patients had to be registered with the practice for a minimum of 12 months prior to cohort entry. Hence, the cohort entry date for each patient was the latest of the study start date (January 1, 2005), practice up-to-standard date, date of 18th birthday, date of registration with the practice plus 12 months, and date of the second eligible (<90 mL/min/1.73 m$^2$) eGFR test. Patient records were censored at the earliest of the study end date (December 31, 2013), date of last upload of practice or linked data, date of death, transfer out of practice date, and date of incident record of pregnancy within study period.

In accordance with the KDIGO classification [14], at cohort entry, participants were categorised according to the value of their second eligible eGFR test as follows: G2 (eGFR 60 to 89 mL/min/1.73 m$^2$); G3a (eGFR 45 to 59 mL/min/1.73 m$^2$); G3b (30 to 44 mL/min/1.73 m$^2$); G4 (eGFR 15 to 29 mL/min/1.73 m$^2$); and G5 (eGFR <15 mL/min/1.73 m$^2$) not on renal replacement therapy (RRT; defined as being with renal transplantation or on maintenance dialysis). The eGFR values were calculated from creatinine using the Chronic Kidney Disease Epidemiology Collaboration (CKD-EPI) equation [18] as recommended by NICE [17].

## The lifetime CKD–CVD policy model

This section contains a brief description of the model development methods. Full details are reported in Methods A in S1 Text. The lifetime chronic kidney disease–cardiovascular disease (CKD–CVD) policy model is a decision-analytic model, consisting of a CKD submodel, which projects kidney function, and a CVD/nonvascular death submodel, which projects the experience of fatal and nonfatal CVD events and nonvascular death (Fig 1). Covariates in the model were defined using the standard clinical terminology system used in the UK general practices, namely the Read Codes and/or other data categories (e.g., entity codes). Cardiovascular events and deaths were defined using both Read and ICD-10 codes (see "**Data Availability Statement**"). Missing

data were imputed using multivariate multiple imputation methods or, as in the case of urinary albumin-to-creatinine ratio (uACR), were assigned into a separate category.

## CVD/nonvascular death submodel

To derive the risk equations, two-thirds of the primary care practices were randomly allocated to the estimation cohort and the remaining one-third to the validation cohort.

The annual risks of 3 nested cardiovascular endpoints: (a) vascular death; (b) vascular death or stroke; and (c) vascular death, stroke, or myocardial infarction (MI) and the risks of nonvascular death and of first hospitalisation for heart failure were estimated (Fig 1). First event of each type was modelled, and events that happened before an event of another type contributed to the estimations (e.g., an MI that occurred before a vascular death would be included as a risk factor in the vascular death risk equation).

For each participant, the risks of these endpoints were estimated using survival risk equations adjusting for a number of baseline characteristics and annually updated age, latest CVD event (including events that occurred during follow-up), and latest eGFR category during follow-up. Additionally, each risk equation included further adjustments for the use of lipid-lowering, antihypertensive, and antiplatelet treatments. Separate risk equations were fitted in participant categories by gender and previous CVD at cohort entry (binary variable: yes/no) to acknowledge different contributions of risk factors. Thus, 4 equations were estimated for each endpoint. Initially, the Andersen–Gill generalisation of the Cox proportional hazards model

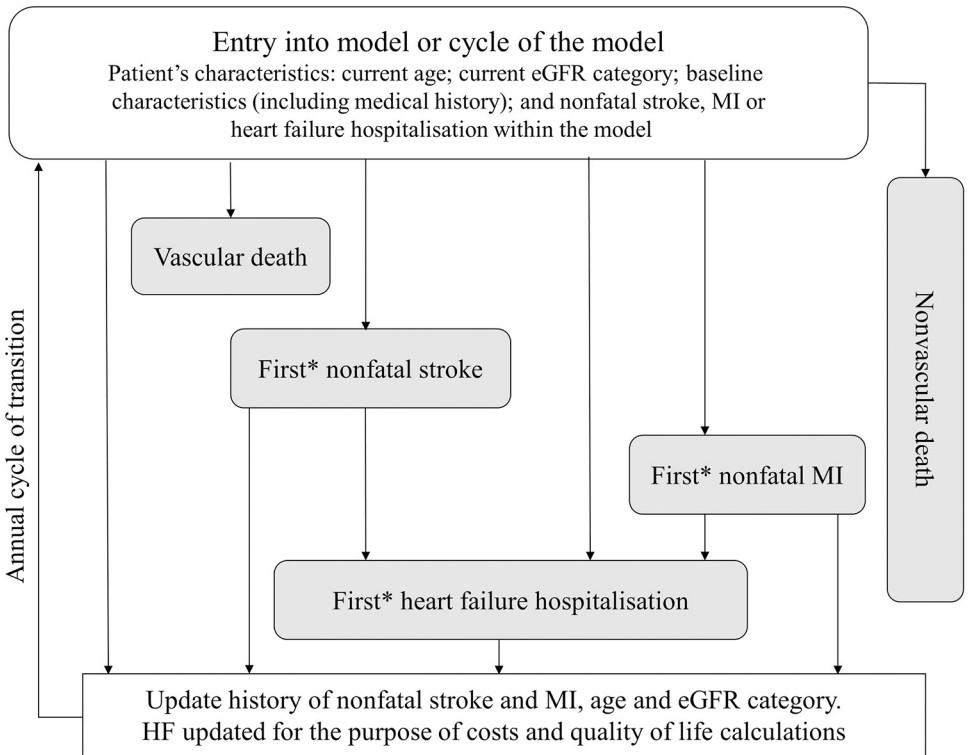

*First within the model

**Fig 1. CVD submodel of the CKD–CVD policy model.** CKD–CVD, chronic kidney disease–cardiovascular disease; CVD, cardiovascular disease; eGFR, estimated glomerular filtration rate; HF, heart failure; MI, myocardial infarction.

was used with all potential covariates included. Age, uACR, kidney disease history, and (for CVD endpoints only) CVD history were retained in all risk equations regardless of their statistical significance. For other covariates, an automatic forward and backwards selection procedure was used. Finally, the variables judged not significant, both statistically (i.e., $p$-value $\geq 0.01$) and clinically (as advised by clinicians), were removed one at a time, and the resulting models were compared to the model without the excluded variable using the likelihood ratio test with $p$-values $<0.01$ deemed statistically significant. Subsequently, parametric proportional hazard survival models, including selected variables, were estimated to support the extrapolation over patient's lifetime. Exponential, Weibull, and Gompertz proportional hazards models were considered, as both external evidence and the Cox modelling confirmed the suitability of the proportional hazards methods, with the choice of the survival distribution guided by minimum Akaike information criterion (AIC)[19].

## CKD submodel

The derivation of the CKD submodel is described in detail elsewhere [20]. Briefly, this submodel simulates progression through renal function stages (G1 [eGFR $\geq 90$ mL/min/1.73 m$^2$]/G2, G3a, G3b, G4, and G5 not on RRT) and death using a hidden Markov model fitted separately to 4 cohorts of patients by baseline albuminuria status (unmeasured, normoalbuminuria, microalbuminuria, and macroalbuminuria [14]). Models were adjusted for patient's sex, diagnoses of heart failure and cancer at cohort entry, and annually updated age. For the purpose of the CKD–CVD policy model, the transition to death was replaced with the estimated risk equations for vascular and nonvascular death as described in the previous section. Additionally, annual transition probabilities from stage G5 not on RRT to dialysis (12% for males $<65$ years old and 5.5% for males $\geq 65$ years old; 9.6% for females $<65$ years old and 2.7% for females $\geq 65$ years old) and renal transplantation (4.3% for males $<65$ years old and 0.1% for males $\geq 65$ years old; 3.7% for females $<65$ years old and 0.1% for females $\geq 65$ years old) were calculated in the CPRD study cohort. Annual rates of renal transplantation while on maintenance dialysis (6.1%) and renal transplantation failure resulting in dialysis initiation (1%) were informed by the UK renal registry data.

## Integrated CKD–CVD model structure

The CVD/nonvascular death and CKD submodels were combined into a Markov model with an annual cycle of transition and transition probabilities between the states derived from the risk equations, as described in the previous sections. The nonfatal model states were defined by the patient's most recent CVD history (stroke, MI in the absence of a stroke, or no event; CVD/nonvascular death submodel) and the latest eGFR category (G2, G3a, G3b, G4, and G5 not on RRT, with renal transplant, on dialysis). Together, the model states consist of all possible combinations of nonfatal CVD events and CKD stages with 2 further fatal states (vascular death and nonvascular death). At start of each annual cycle, patient's age, CKD stage, and CVD event status are updated. The information then initially feeds into the CKD submodel, in which patient's next CKD stage is projected, and then into the CVD/nonvascular death submodel, in which patient's CVD status is projected. The model then enters the next annual cycle, where the updated disease histories and age are used.

 The risk equation for heart failure was used in the model to project the impact of heart failure on health-related quality of life and healthcare costs but not on the subsequent CVD events.

## Health-related quality of life and healthcare costs

Health-related quality of life utilities and annual healthcare costs, corresponding to each model state, were informed by external data (Tables A and B in S1 Text). Annual healthcare

costs corresponding to earlier years were inflated to year 2017 using the hospital and community health services index.

## Uncertainty

Parameter uncertainty in the model was propagated by sampling all parameters from predefined probability distributions (multivariate normal distribution for the CVD and CKD risk equations, binomial distribution for transitions into, and between, renal transplantation and dialysis, gamma distribution for the cost equation and normal distribution for the quality of life equation), generating 1,000 sets of risk, cost, and quality of life equations and simulating the results with these parameter sets. Confidence intervals (CIs) were then derived using the equal-tailed percentile method.

## Model validation

Model-simulated cumulative rates of cardiovascular endpoints were validated by comparing the simulated cumulative rates with the Kaplan–Meier product limit estimates over 5 years of follow-up among patients in the validation cohort, by eGFR category at cohort entry. Model discrimination was assessed using Harrell's C-index for censored response. Model-simulated cumulative rates of vascular death, stroke, or MI were also compared with the respective simulated Kaplan–Meier product limit estimates of cumulative rates over 5 years of follow-up separately in each of the 10 regions of England, by eGFR category at cohort entry.

## Policy applications

The CKD–CVD policy model was used to simulate the remaining life expectancy of people with reduced kidney function and quantify the contribution of widely used treatments to reduce cardiovascular risk, namely statins, antihypertensives, and antiplatelets. For these analyses, a random sample of 64,000 patients was sampled from the CPRD cohort. The sampling was performed separately in groups of patients by gender, previous CVD, and eGFR category (5,000 each from the categories G2, G3a, and G3b and 1,000 from the combined G4 and G5 not on RRT category) maintaining the age distribution within each group.

**Application 1: Predicting remaining life expectancy of people with reduced kidney function.** Firstly, the CKD–CVD model was used to predict remaining life expectancy and quality-adjusted life expectancy for the sampled cohort using patients' characteristics at entry into the cohort and assuming no use of statins, antihypertensives, or antiplatelets.

**Application 2: Quantifying the impact of partial and optimal use of cardiovascular prevention medications in people with reduced kidney function.** Secondly, the CKD–CVD policy model was used to project (a) the (quality-adjusted) life expectancy gained with 2013 levels of use of statins, antihypertensives, and antiplatelets (i.e. partial use) compared to no use; and (b) the additional (quality-adjusted) life years that could be gained with optimal guideline-indicated use of these cardioprotective medications [17,21,22] compared to partial use. The effects of these treatments (relative risks) on cardiovascular event risks in the model were informed from recent meta-analyses of randomised trials [23–25] (Methods A in S1 Text).

## Software

All analyses were performed using R or Stata 15 [26,27]. The figures were produced using the R ggplot2 package [28]; see also "**Data Availability Statement**".

### Ethics statement

The protocol for this research was approved by the Independent Scientific Advisory Committee (ISAC) of the Medicines and Healthcare Products Regulatory Agency (protocol number 14_150RA, available on request). Ethical approval for observational research using the CPRD with approval from ISAC has been granted by a National Research Ethics Service committee (Trent MultiResearch Ethics Committee, REC reference number 05/MRE04/87). No consent was required as the data were analysed anonymously.

## Results

The study cohort consisted of 1,139,548 patients with reduced kidney function (Fig A in S1 Text), with a median follow-up of 4.9 years (interquartile range 2.3 to 7.8 years). At entry, the vast majority of participants were in eGFR categories G2 (74%) or G3a (18%), with only a few patients in G3b (6%), G4 (1.3%), or G5 not on RRT (0.2%). The baseline characteristics of participants in the estimation (Table C in S1 Text) and validation (Table D in S1 Text) cohorts were broadly similar and confirm increased prevalence of a range of vascular and other comorbidities with worsening kidney function. Participants in G5 not on RRT at entry were somewhat younger than other participants with moderate to severe renal impairment, e.g., in the estimation cohort, the mean age at cohort entry was 71 years among patients in G5 not on RRT and 81 years among patients in G4 (72 and 81 years, respectively, in the validation cohort).

During follow-up, larger proportions of participants with greater kidney function impairment experienced cardiovascular events or died (Table E in S1 Text). For example, in the estimation cohort, among the participants in eGFR category G2 at cohort entry, 2% died from cardiovascular causes, and 5% experienced the combined endpoint of vascular death, stroke, or MI compared with 12% and 22%, respectively, among participants in G5 not on RRT at cohort entry. Details of transitions between eGFR categories have been published elsewhere [20].

### Risk equations

The risk equations for the cardiovascular endpoints (vascular death, vascular death or stroke, vascular death, stroke, or MI) indicated that more severe eGFR categories were associated with increased cardiovascular risk, with clearly graded associations observed across all eGFR categories of patients (Tables 1 and 2 for females and males, respectively). A stroke or an MI during follow-up also substantially increased the subsequent risk of cardiovascular death, with stronger proportional increases among patients without previous CVD. Age, type 1 diabetes, smoking, and history of CVD at cohort entry were also associated with substantially increased cardiovascular risks. A graded association was observed between worsening eGFR category and higher risks of nonvascular death (Table F in S1 Text) and heart failure hospital admission (Table G in S1 Text).

### Model validation

The cumulative incidence of cardiovascular events simulated by the model across the 5 years of follow-up closely matched the observed rates for most years and eGFR categories of patients in the validation cohort (Fig 2; Fig B in S1 Text) as well as across the 10 regions in England (Fig C in S1 Text). The Harrell's C-index, ranging from 0.60 to 0.84 in the estimation cohort and from 0.62 to 0.84 in the validation cohort, indicated that the risks of key disease events were well discriminated across the eGFR categories (Table H in S1 Text).

**Table 1. Risk equations for CVD endpoints in the CKD–CVD policy model (females).**

| Covariates | Category | Females with no previous CVD | | | Females with previous CVD | | |
|---|---|---|---|---|---|---|---|
| | | Vascular death | Vascular death or stroke | Vascular death, stroke, or MI | Vascular death | Vascular death or stroke | Vascular death, stroke, or MI |
| | | Gompertz PH HR (95% CI) | Weibull PH HR (95% CI) | Weibull PH HR (95% CI) | Gompertz PH HR (95% CI) | Weibull PH HR (95% CI) | Weibull PH HR (95% CI) |
| **Patient's characteristics at cohort entry** | | | | | | | |
| *Sociodemographic characteristics* | | | | | | | |
| Smoker (ref: never smoked) | Former | 1.11 (1.06, 1.18) | 1.08 (1.04, 1.13) | 1.10 (1.07, 1.14) | 1.02 (0.96, 1.09) | 0.98 (0.93, 1.03) | 1.02 (0.98, 1.07) |
| | Current | 1.71 (1.60, 1.83) | 1.70 (1.63, 1.78) | 1.81 (1.74, 1.88) | 1.32 (1.21, 1.45) | 1.33 (1.24, 1.42) | 1.40 (1.32, 1.48) |
| BMI (ref: $\geq$18.5, <25 kg/m$^2$) | <18.5 kg/m$^2$ | 1.43 (1.27, 1.61) | 1.31 (1.19, 1.45) | 1.30 (1.19, 1.41) | 1.42 (1.27, 1.58) | 1.33 (1.21, 1.46) | 1.35 (1.23, 1.47) |
| | $\geq$25, <30 kg/m$^2$ | 0.88 (0.84, 0.93) | 0.92 (0.88, 0.95) | 0.94 (0.91, 0.97) | 0.84 (0.78, 0.91) | 0.88 (0.82, 0.94) | 0.88 (0.83, 0.93) |
| | $\geq$30, <35 kg/m$^2$ | 0.88 (0.82, 0.95) | 0.95 (0.90, 1.00) | 0.95 (0.91, 0.99) | 0.78 (0.72, 0.85) | 0.84 (0.79, 0.89) | 0.86 (0.81, 0.91) |
| | $\geq$35, <40 kg/m$^2$ | 1.16 (1.03, 1.31) | 1.06 (0.98, 1.14) | 1.04 (0.97, 1.11) | 0.87 (0.74, 1.02) | 0.93 (0.83, 1.04) | 0.93 (0.85, 1.03) |
| | $\geq$40 kg/m$^2$ | 1.78 (1.53, 2.08) | 1.41 (1.27, 1.57) | 1.25 (1.14, 1.37) | 1.18 (1.08, 1.28) | 0.91 (0.79, 1.04) | 0.87 (0.77, 0.98) |
| Index of multiple deprivation quintile (ref: First quintile: least deprived) | Second quintile | 1.06 (0.99, 1.13) | 1.04 (1.00, 1.10) | 1.05 (1.00, 1.09) | 1.07 (0.99, 1.15) | 1.02 (0.96, 1.09) | 1.02 (0.97, 1.08) |
| | Third quintile | 1.25 (1.17, 1.33) | 1.16 (1.11, 1.22) | 1.17 (1.12, 1.22) | 1.06 (0.98, 1.15) | 1.05 (0.99, 1.12) | 1.05 (1.00, 1.11) |
| | Fourth quintile | 1.20 (1.12, 1.28) | 1.15 (1.09, 1.21) | 1.15 (1.10, 1.20) | 1.09 (1.00, 1.18) | 1.04 (0.98, 1.11) | 1.07 (1.01, 1.13) |
| | Fifth quintile (most deprived) | 1.27 (1.17, 1.36) | 1.23 (1.17, 1.30) | 1.25 (1.19, 1.31) | 1.18 (1.08, 1.28) | 1.16 (1.09, 1.24) | 1.18 (1.11, 1.25) |
| *Disease history, laboratory measurements, and other risk factors* | | | | | | | |
| Diabetes (ref: no diabetes) | Type I | 4.70 (3.05, 7.23) | 3.66 (2.74, 4.91) | 4.40 (3.52, 5.51) | 4.39 (2.95, 6.53) | 3.45 (2.55, 4.68) | 3.50 (2.71, 4.53) |
| | Type II | 1.45 (1.35, 1.55) | 1.37 (1.30, 1.44) | 1.39 (1.33, 1.46) | 1.44 (1.34, 1.55) | 1.40 (1.32, 1.48) | 1.41 (1.34, 1.48) |
| Albuminuria status (ref: not measured) | Normoalbuminuria | 0.78 (0.75, 0.82) | 0.86 (0.84, 0.89) | 0.89 (0.86, 0.91) | 0.78 (0.74, 0.83) | 0.84 (0.80, 0.87) | 0.88 (0.85, 0.91) |
| | Microalbuminuria | 0.99 (0.85, 1.15) | 1.11 (0.99, 1.24) | 1.10 (1.00, 1.22) | 0.89 (0.77, 1.03) | 0.94 (0.83, 1.05) | 1.00 (0.90, 1.10) |
| | Macroalbuminuria | 1.25 (0.98, 1.59) | 1.37 (1.30, 1.44) | 1.41 (1.20, 1.67) | 1.36 (1.10, 1.67) | 1.38 (1.16, 1.64) | 1.34 (1.14, 1.57) |
| Total cholesterol: HDL cholesterol ratio | Per unit increase | 1.01 (0.99, 1.02) | 1.00 (0.99, 1.02) | 1.02 (1.01, 1.02) | 1.01 (0.98, 1.04) | 0.99 (0.97, 1.01) | 1.03 (1.01, 1.05) |
| Systolic blood pressure centred at 139 mmHg | Per 20 mmHg increase | 1.00 (0.98, 1.02) | 1.05 (1.03, 1.07) | 1.07 (1.06, 1.09) | 0.93 (0.91, 0.96) | 1.00 (0.98, 1.02) | 1.02 (1.00, 1.05) |
| Rheumatoid arthritis | Yes | 1.48 (1.31, 1.66) | 1.40 (1.28, 1.53) | 1.43 (1.32, 1.54) | 1.32 (1.16, 1.50) | 1.24 (1.12, 1.38) | 1.24 (1.13, 1.36) |
| Atrial fibrillation | Yes | 1.81 (1.67, 1.97) | 1.80 (1.68, 1.92) | 1.64 (1.54, 1.75) | 1.34 (1.26, 1.42) | 1.38 (1.32, 1.45) | 1.18 (1.13, 1.24) |
| Diagnosis of mental illness | Yes | 1.39 (1.27, 1.53) | 1.29 (1.21, 1.39) | 1.29 (1.21, 1.37) | 1.27 (1.15, 1.42) | 1.29 (1.19, 1.40) | 1.24 (1.15, 1.34) |
| Family history of coronary heart disease | Yes | 0.92 (0.86, 0.98) | 0.95 (0.91, 1.00) | 1.03 (0.99, 1.08) | N/A | | |

(*Continued*)

**Table 1.** (Continued)

| Covariates | Category | Females with no previous CVD | | | Females with previous CVD | | |
|---|---|---|---|---|---|---|---|
| | | Vascular death | Vascular death or stroke | Vascular death, stroke, or MI | Vascular death | Vascular death or stroke | Vascular death, stroke, or MI |
| | | Gompertz PH HR (95% CI) | Weibull PH HR (95% CI) | Weibull PH HR (95% CI) | Gompertz PH HR (95% CI) | Weibull PH HR (95% CI) | Weibull PH HR (95% CI) |
| History of coronary heart disease | Yes | N/A | | | 1.24 (1.16, 1.32) | 1.21 (1.15, 1.28) | 1.37 (1.31, 1.44) |
| History of cerebrovascular disease | Yes | | | | 1.64 (1.55, 1.74) | 1.70 (1.62, 1.78) | 1.54 (1.48, 1.61) |
| History of heart failure | Yes | | | | 1.60 (1.51, 1.70) | 1.42 (1.35, 1.19) | 1.40 (1.33, 1.46) |
| **Characteristics updated on an annual basis** | | | | | | | |
| Age | Per 10 years older | 3.32 (3.23, 3.41) | 2.50 (2.45, 2.54) | 2.31 (2.27, 2.35) | 2.33 (2.26, 2.41) | 1.91 (1.86, 1.96) | 1.79 (1.75, 1.83) |
| eGFR category at the end of the previous year (ref: eGFR 60–89 mL/min/1.73 m$^2$ [G2]) | eGFR 45–59 mL/min/1.73 m$^2$ (G3a) | 1.13 (1.07, 1.18) | 1.12 (1.07, 1.16) | 1.11 (1.07, 1.14) | 1.07 (1.00, 1.14) | 1.06 (1.04, 1.15) | 1.11 (1.06, 1.73) |
| | eGFR 30–44 mL/min/1.73 m$^2$ (G3b) | 1.36 (1.28, 1.45) | 1.27 (1.21, 1.33) | 1.30 (1.25, 1.36) | 1.36 (1.27, 1.46) | 1.31 (1.24, 1.39) | 1.34 (1.27, 1.41) |
| | eGFR 15–29 mL/min/1.73 m$^2$ (G4) | 2.11 (1.93, 2.30) | 1.87 (1.74, 2.00) | 1.88 (1.76, 2.01) | 1.79 (1.64, 1.95) | 1.63 (1.51, 1.76) | 1.61 (1.50, 1.73) |
| | eGFR <15 mL/min/1.73 m$^2$ (G5) or RRT | 3.99 (3.15, 5.05) | 3.40 (2.79, 4.14) | 3.46 (2.89, 4.14) | 2.75 (2.20, 3.43) | 2.16 (1.77, 2.63) | 2.33 (1.96, 2.77) |
| Cardiovascular event during follow-up (ref: no MI or stroke during follow-up) | MI | 2.14 (1.88, 2.45) | 1.89 (1.70, 2.09) | N/A | 1.67 (1.45, 1.92) | 1.52 (1.35, 1.70) | N/A |
| | Stroke | 3.56 (3.27, 3.87) | N/A | | 2.59 (2.35, 2.86) | N/A | |
| Intercept | | −7.226 (−7.330, −7.122) | −5.938 (−6.017, −5.859) | −5.659 (−5.714, −5.605) | −5.869 (−6.025, −5.714) | −4.932 (−5.052, −4.812) | −4.780 (−4.895, −4.665) |
| Ancillary parameter | | −0.027 (−0.035, −0.018) | −0.060 (−0.074, −0.045) | −0.049 (−0.062, −0.036) | −0.049 (−0.060, −0.039) | −0.103 (−0.120, −0.086) | −0.097 (−0.112, −0.081) |

BMI, body mass index; CKD, chronic kidney disease; CKD–CVD, chronic kidney disease–cardiovascular disease; CVD, cardiovascular disease; eGFR, estimated glomerular filtration rate; HDL, high-density lipoprotein; HR, hazard ratio; MI, Myocardial infarction; N/A, not applicable: covariate not included or specified through other covariates within category; PH, proportional hazards; RRT, renal replacement therapy.

Each risk equation included further adjustments for use of lipid-lowering, antihypertensive, and antiplatelet therapies. The intercept and ancillary parameters are presented on the original scale.

**Application 1: Predicting life expectancy of patients with reduced kidney function.** The model predicted large variation in survival by eGFR category at cohort entry (Fig 3, Table I in S1 Text). Among patients aged 60 to 69 in G2, those without previous CVD were predicted to live a further 22.1 years (16.1 quality-adjusted life years [QALYs]) and those with CVD a further 18.6 years (11.8 QALYs). The corresponding values for patients in G5 not on RRT were 13.7 years (9.2 QALYs) and 10.1 years (5.7 QALYs), respectively.

**Application 2: Quantifying the impact of partial and optimal use of cardiovascular prevention medications in patients with reduced kidney function.** Fig 4 and Table J in in S1 Text summarise the benefits of statins, antihypertensives, and antiplatelets by patient's age, eGFR category, and history of CVD at cohort entry. Using NICE recommendations in 2019, the proportions of patients indicated for treatment who were prescribed statin, antihypertensive, or antiplatelet in 2013 would be 37% among those without previous CVD and 71%

**Table 2. Risk equations for CVD endpoints in the CKD–CVD policy model (males).**

| Covariates | Category | Males without previous CVD | | | Males with previous CVD | | |
|---|---|---|---|---|---|---|---|
| | | Vascular death | Vascular death or stroke | Vascular death, stroke, or MI | Vascular death | Vascular death or stroke | Vascular death, stroke, or MI |
| | | Gompertz PH HR (95% CI) | Weibull PH HR (95% CI) | Weibull PH HR (95% CI) | Gompertz PH HR (95% CI) | Weibull PH HR (95% CI) | Weibull PH HR (95% CI) |
| **Patient's characteristics at cohort entry** | | | | | | | |
| *Sociodemographic characteristics* | | | | | | | |
| Smoker (ref: never smoked) | Former | 1.12 (1.06, 1.19) | 1.08 (1.03, 1.12) | 1.09 (1.06, 1.13) | 1.02 (0.96, 1.08) | 1.00 (0.96, 1.04) | 1.03 (0.99, 1.07) |
| | Current | 1.79 (1.67, 1.91) | 1.65 (1.57, 1.74) | 1.65 (1.59, 1.72) | 1.30 (1.20, 1.40) | 1.25 (1.18, 1.33) | 1.30 (1.24, 1.37) |
| BMI (ref: $\geq$18.5, <25 kg/m$^2$) | <18.5 kg/m$^2$ | 1.57 (1.23, 2.00) | 1.53 (1.30, 1.79) | 1.35 (1.16, 1.57) | 1.76 (1.38, 2.26) | 1.55 (1.28, 1.87) | 1.42 (1.19, 1.69) |
| | $\geq$25, <30 kg/m$^2$ | 0.89 (0.83, 0.95) | 0.93 (0.88, 0.97) | 0.94 (0.91, 0.98) | 0.84 (0.79, 0.89) | 0.89 (0.85, 0.94) | 0.90 (0.87, 0.94) |
| | $\geq$30, <35 kg/m$^2$ | 0.98 (0.89, 1.07) | 0.96 (0.91, 1.03) | 1.00 (0.95, 1.05) | 0.83 (0.76, 0.90) | 0.88 (0.83, 0.94) | 0.87 (0.82, 0.92) |
| | $\geq$35, <40 kg/m$^2$ | 1.30 (1.13, 1.51) | 1.15 (1.04, 1.27) | 1.06 (0.98, 1.16) | 0.86 (0.72, 1.01) | 0.84 (0.73, 0.95) | 0.80 (0.72, 0.89) |
| | $\geq$40 kg/m$^2$ | 1.68 (1.34, 2.09) | 1.45 (1.25, 1.69) | 1.27 (1.12, 1.44) | 1.45 (1.19, 1.78) | 1.21 (1.01, 1.44) | 1.04 (0.89, 1.21) |
| Index of multiple deprivation quintile (ref: First quintile: least deprived) | Second quintile | 1.11 (1.03, 1.19) | 1.11 (1.06, 1.17) | 1.11 (1.07, 1.16) | 1.06 (0.98, 1.15) | 1.04 (0.98, 1.11) | 1.09 (1.03, 1.15) |
| | Third quintile | 1.17 (1.08, 1.26) | 1.20 (1.14, 1.26) | 1.19 (1.14, 1.24) | 1.16 (1.07, 1.26) | 1.10 (1.03, 1.17) | 1.12 (1.06, 1.18) |
| | Fourth quintile | 1.18 (1.09, 1.28) | 1.19 (1.13, 1.26) | 1.18 (1.13, 1.23) | 1.20 (1.11, 1.30) | 1.18 (1.11, 1.26) | 1.20 (1.14, 1.27) |
| | Fifth quintile (most deprived) | 1.39 (1.27, 1.51) | 1.36 (1.28, 1.45) | 1.32 (1.26, 1.39) | 1.33 (1.22, 1.45) | 1.26 (1.18, 1.35) | 1.30 (1.23, 1.37) |
| *Disease history, laboratory measurements, and other risk factors* | | | | | | | |
| Diabetes (ref: no diabetes) | Type I | 2.40 (1.50, 3.83) | 2.17 (1.61, 2.92) | 2.26 (1.81, 2.82) | 3.09 (2.25, 4.24) | 2.56 (2.00, 3.26) | 2.33 (1.90, 2.85) |
| | Type II | 1.14 (1.06, 1.23) | 1.18 (1.12, 1.24) | 1.19 (1.14, 1.24) | 1.45 (1.36, 1.56) | 1.40 (1.33, 1.48) | 1.39 (1.33, 1.45) |
| Albuminuria status (ref: not measured) | Normoalbuminuria | 0.89 (0.84, 0.94) | 0.93 (0.89, 0.96) | 0.94 (0.92, 0.97) | 0.82 (0.77, 0.86) | 0.87 (0.83, 0.90) | 0.91 (0.87, 0.94) |
| | Microalbuminuria | 1.18 (1.01, 1.37) | 1.15 (1.03, 1.28) | 1.18 (1.08, 1.30) | 0.90 (0.79, 1.02) | 1.00 (0.90, 1.11) | 1.06 (0.97, 1.16) |
| | Macroalbuminuria | 1.28 (1.00, 1.62) | 1.37 (1.14, 1.64) | 1.36 (1.17, 1.59) | 1.00 (0.82, 1.20) | 1.00 (0.85, 1.17) | 1.09 (0.95, 1.24) |
| Total cholesterol: HDL cholesterol ratio | Per unit increase | 1.02 (0.99, 1.04) | 1.01 (1.00, 1.03) | 1.06 (1.05, 1.07) | 1.02 (1.00, 1.03) | 1.01 (0.99, 1.02) | 1.03 (1.02, 1.05) |
| Systolic blood pressure centred at 139 mmHg | Per 20 mmHg increase | 1.02 (0.99, 1.05) | 1.08 (1.05, 1.10) | 1.08 (1.06, 1.10) | 0.96 (0.94, 0.99) | 1.03 (1.01, 1.05) | 1.02 (1.00, 1.04) |
| Rheumatoid arthritis | Yes | 1.42 (1.16, 1.74) | 1.32 (1.14, 1.53) | 1.33 (1.18, 1.50) | 1.56 (1.31, 1.85) | 1.44 (1.25, 1.66) | 1.39 (1.23, 1.57) |
| Atrial fibrillation | Yes | 1.65 (1.51, 1.81) | 1.57 (1.46, 1.69) | 1.42 (1.33, 1.51) | 1.25 (1.17, 1.33) | 1.27 (1.21, 1.34) | 1.04 (1.00, 1.09) |
| Diagnosis of mental illness | Yes | 1.62 (1.43, 1.85) | 1.45 (1.32, 1.59) | 1.28 (1.18, 1.38) | 1.40 (1.22, 1.59) | 1.31 (1.18, 1.45) | 1.24 (1.14, 1.36) |
| Family history of coronary heart disease | Yes | 0.97 (0.89, 1.05) | 0.96 (0.91, 1.01) | 1.13 (1.09, 1.18) | N/A | | |

(*Continued*)

**Table 2.** (Continued)

| Covariates | Category | Males without previous CVD | | | Males with previous CVD | | |
|---|---|---|---|---|---|---|---|
| | | Vascular death | Vascular death or stroke | Vascular death, stroke, or MI | Vascular death | Vascular death or stroke | Vascular death, stroke, or MI |
| | | Gompertz PH HR (95% CI) | Weibull PH HR (95% CI) | Weibull PH HR (95% CI) | Gompertz PH HR (95% CI) | Weibull PH HR (95% CI) | Weibull PH HR (95% CI) |
| History of coronary heart disease | Yes | N/A | | | 1.13 (1.06, 1.22) | 1.06 (1.01, 1.12) | 1.54 (1.48, 1.60) |
| History of cerebrovascular disease | Yes | | | | 1.69 (1.59, 1.79) | 1.75 (1.67, 1.83) | 1.41 (1.35, 1.47) |
| History of heart failure | Yes | | | | 1.71 (1.61, 1.82) | 1.43 (1.36, 1.51) | 1.26 (1.20, 1.32) |
| **Characteristics updated on an annual basis** | | | | | | | |
| Age | Per 10 years older | 2.73 (2.65, 2.82) | 2.21 (2.16, 2.26) | 1.92 (1.89, 1.96) | 2.11 (2.04, 2.18) | 1.77 (1.73, 1.82) | 1.60 (1.56, 1.63) |
| eGFR category at the end of the previous year (ref: eGFR 60–89 mL/min/1.73 m$^2$[G2]) | eGFR 45–59 mL/min/1.73 m$^2$ (G3a) | 1.27 (1.19, 1.35) | 1.22 (1.16, 1.27) | 1.21 (1.16, 1.25) | 1.10 (1.03, 1.17) | 1.08 (1.03, 1.14) | 1.07 (1.03, 1.12) |
| | eGFR 30–44 mL/min/1.73 m$^2$ (G3b) | 1.83 (1.70, 1.98) | 1.53 (1.44, 1.62) | 1.52 (1.45, 1.60) | 1.57 (1.46, 1.69) | 1.44 (1.36, 1.52) | 1.38 (1.31, 1.45) |
| | eGFR 15–29 mL/min/1.73 m$^2$ (G4) | 2.52 (2.25, 2.82) | 1.99 (1.81, 2.18) | 1.92 (1.77, 2.09) | 2.17 (1.97, 2.39) | 1.81 (1.67, 1.97) | 1.76 (1.63, 1.89) |
| | eGFR <15 mL/min/1.73 m$^2$ (G5) or RRT | 3.86 (3.05, 4.88) | 3.02 (2.49, 3.67) | 3.16 (2.69, 3.73) | 3.13 (2.62, 3.75) | 2.55 (2.18, 2.99) | 2.60 (2.27, 2.98) |
| Cardiovascular event during follow-up (ref: no MI or stroke during follow-up) | MI | 2.05 (1.80, 2.34) | 1.57 (1.41, 1.74) | N/A | 1.90 (1.69, 2.14) | 1.67 (1.52, 1.84) | N/A |
| | Stroke | 3.42 (3.10, 3.77) | N/A | | 2.40 (2.17, 2.66) | N/A | |
| Intercept | | −6.852 (−6.988, −6.716) | −5.672 (−5.760, −5.584) | −5.247 (−5.306, −5.188) | −5.722 (−5.865, −5.579) | −4.707 (−4.822, −4.591) | −4.404 (−4.496, −4.313) |
| Ancillary parameter | | −0.038 (−0.048, −0.028) | −0.065 (−0.082, −0.049) | −0.071 (−0.084, −0.058) | −0.040 (−0.051, −0.029) | −0.097 (−0.114, −0.079) | −0.101 (−0.116, −0.087) |

BMI, body mass index; CKD, chronic kidney disease; CKD–CVD, chronic kidney disease–cardiovascular disease; CVD, cardiovascular disease; eGFR, estimated glomerular filtration rate; HDL, high-density lipoprotein; HR, hazard ratio; MI, Myocardial infarction; N/A, not applicable: covariate not included or specified through other covariates within category; PH, proportional hazards; RRT, renal replacement therapy.

Each risk equation included further adjustments for use of lipid-lowering, antihypertensive, and antiplatelet therapies. The intercept and ancillary parameters are presented on the original scale.

among those with previous CVD. The predicted benefits of this level of treatment are substantial across all patient categories, with younger patients with previous CVD deriving largest benefits. Among patients aged 60 to 69 without previous CVD, the use of these treatments at 2013 levels is evaluated to add a further 0.19 years (0.22 QALYs) to the life expectancy of those in G2, 0.20 years (0.22 QALYs) in G3a, 0.22 years (0.22 QALYs) in G3b, 0.25 years (0.23 QALYs) in G4, and 0.22 years (0.21 QALYs) in G5 not on RRT. For patients aged 60 to 69 with previous CVD, the gains in life expectancy are higher at 0.90 years (0.80 QALYs) in G2, 0.92 years (0.82 QALYs) in G3a, 1.01 years (0.81 QALYs) in G3b, 1 year (0.78 QALYs) in G4, and 0.95 years (0.72 QALYs) in G5 not on RRT.

Increasing treatment use to the optimal guideline-indicated levels in 2019 would add further 0.33 to 0.43 years (0.35 to 0.39 QALYs) in patients 60 to 69 years old without previous CVD at cohort entry and between 0.37 and 0.41 years (0.30 and 0.34 QALYs) in patients 60 to 69 years with previous CVD at cohort entry.

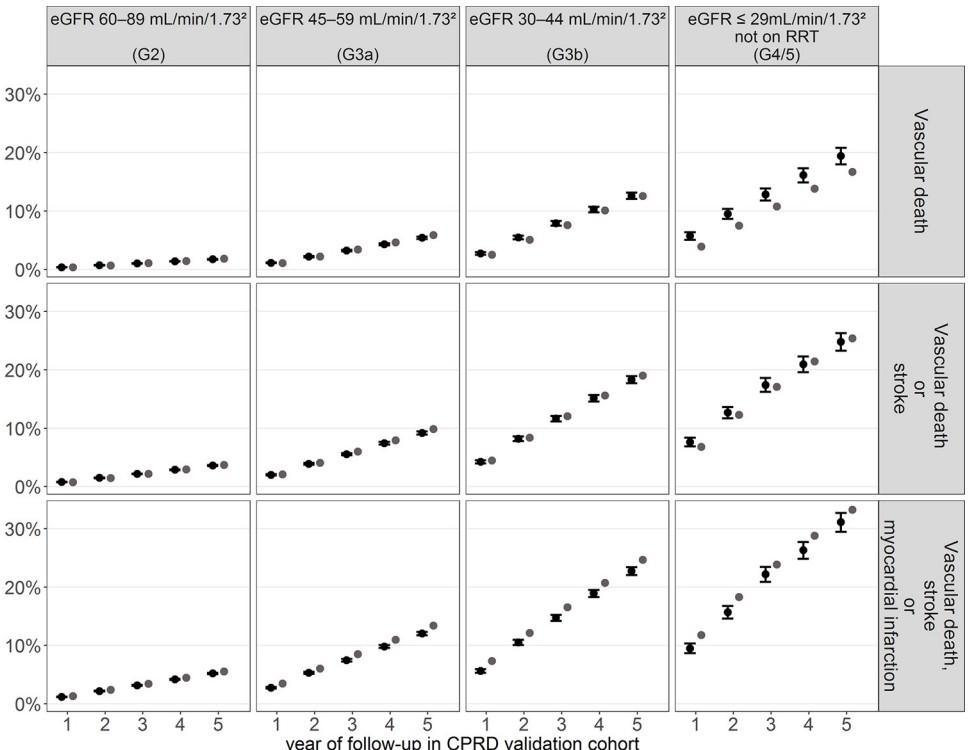

**Fig 2. Comparison of cumulative risks predicted by the model and observed Kaplan–Meier risks in the validation cohort for main cardiovascular endpoints and by eGFR category at cohort entry.** CI, confidence interval; CKD, chronic kidney disease; CPRD, Clinical Practice Research Datalink; CVD, cardiovascular disease; eGFR, estimated glomerular filtration rate; RRT, renal replacement therapy.

## Discussion

In the current study, we present a novel policy model of kidney disease progression and cardiovascular complications in people with reduced kidney function and quantify the benefits of cardioprotective medications in this population. The model demonstrates good disease risk discrimination and predictive accuracy and can be used to project the decline in patients' kidney function, experience of cardiovascular events, healthcare costs, quality of life, and survival using patients' characteristics at entry. To illustrate the model's potential to inform policy, we evaluated the gains in (quality-adjusted) survival achieved with partial use of cardioprotective treatments in patients with reduced kidney function (i.e., at 2013 levels), as well as further gains that could be achieved with optimised treatments according to current guidelines.

The graded relationships between more severe kidney function impairment and increased CVD risk and between the experience of cardiovascular events and cardiovascular mortality are consistent with reports from other patient data [5]. These associations, estimated separately in patient categories by sex and previous CVD, are stronger among men and among patients without history of CVD. This difference in strengths of associations has been previously reported with respect to effect of cardiovascular events on subsequent cardiovascular risk [29,30] but is less well studied with respect to the impact of kidney function impairment on cardiovascular risk, with data typically presented across both sexes [3,31].

The predicted life expectancies using the CKD–CVD policy model were somewhat larger than the predictions using a previous model, which was informed by data from moderate to severe CKD patients recruited in hospitals [5], which indicates that the population with

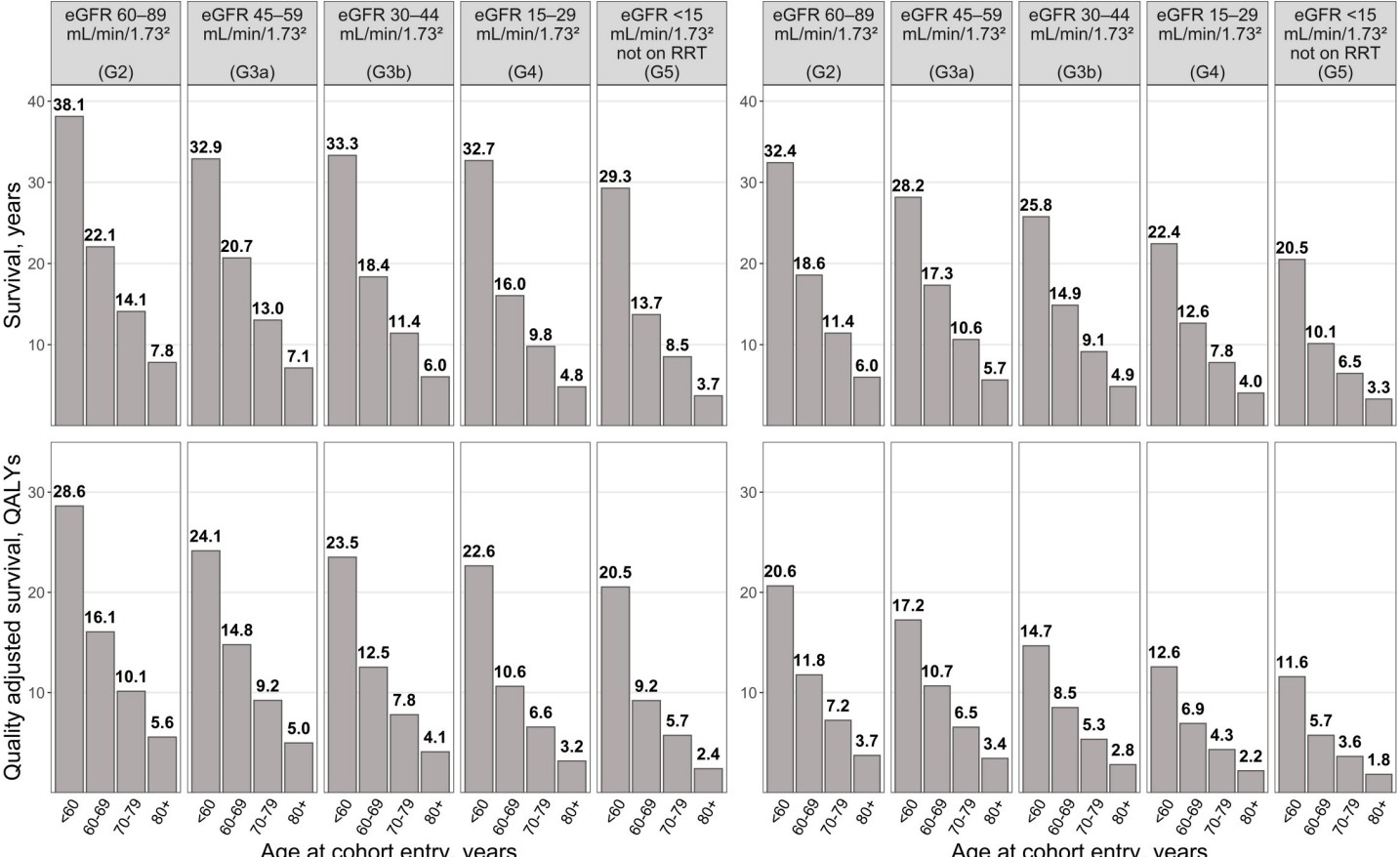

**Fig 3. Predicted life expectancy and QALYs in the absence of cardiovascular prevention treatments by CVD history, age, and eGFR category at cohort entry.** CKD–CVD, chronic kidney disease–cardiovascular disease; CVD, cardiovascular disease; eGFR, estimated glomerular filtration rate; QALY, quality-adjusted life year; RRT, renal replacement therapy. The predictions by the CKD–CVD model are based on a random sample of 64,000 patients from the whole cohort; see Methods section for further detail on how the sampling was performed.

reduced kidney function seen in primary care differs from the CKD patients seen in secondary care. The CKD–CVD model allowed us to estimate for the first time the population gains in life expectancy from the use of 3 commonly indicated cardiovascular prevention treatments (statins, antihypertensives, and antiplatelets) in UK primary care, including the potential for further gains with the optimised use of these treatments. These projected gains in survival are consistent with previous analyses of effects of individual interventions such as statins in CKD.

The major strength of our model is that it is based on a large open cohort of unselected population of patients with reduced kidney function from 388 UK primary care practices, with available information for a wide range of individual patient characteristics. This has helped to overcome a number of limitations of previously published CKD models. It has assured model generalisability and allowed assessment of disease risks across individuals with different disease risks, including people with mild kidney impairment (i.e., prior to the onset of clinical CKD), which constituted a large part of the cohort. Furthermore, the model demonstrated good validity, with good discrimination and calibration across eGFR categories and geographical regions in the cohort.

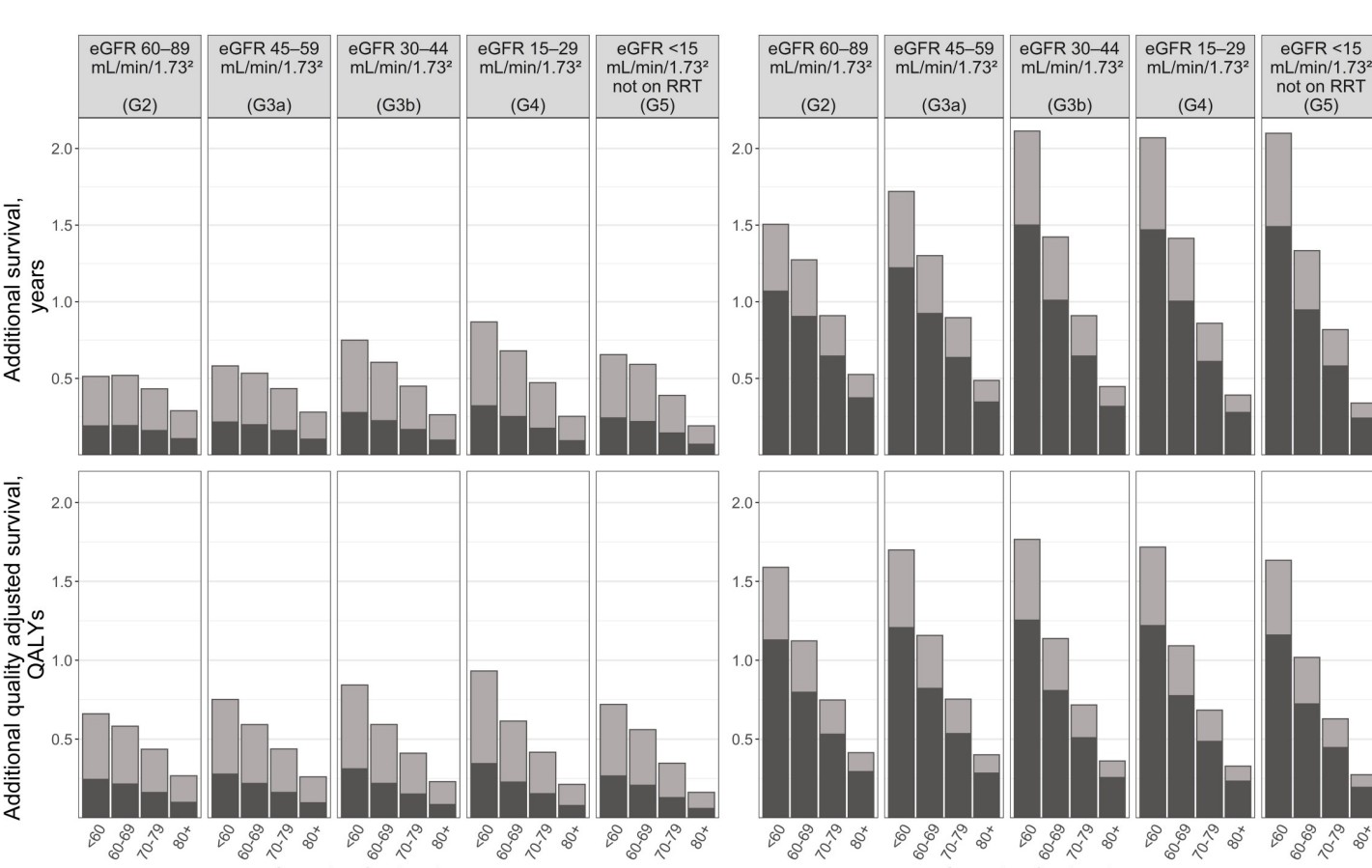

**Fig 4.** Predicted life years and QALYs gained with cardiovascular prevention medications as per (a) use in 2013 compared to no use (black bars) and additional gains with (b) optimal guideline-indicated use in 2019 (grey bars) by CVD history, age, and eGFR category at cohort entry. CVD, cardiovascular disease; eGFR, estimated glomerular filtration rate; NICE, National Institute for Health and Care Excellence; QALY, quality-adjusted life year; RRT, renal replacement therapy. The predictions are based on a random sample of 64,000 patients from the whole cohort. Of the patients indicated for particular cardiovascular prevention treatment in 2019, 37% of those without CVD and 71% of those with CVD were projected to have been treated in 2013. These proportions were calculated using data on patients that were in the study cohort in 2013 and the 2019 UK NICE guidelines. See Methods and S1 Methods in S1 Text for more information.

There are a number of limitations to the model, mostly related to the use of routine data. In routine healthcare practice, risk factors are measured at points of contact with health services, and the purpose of data measurement and recording is to support clinical care of patients and operations of healthcare providers. Therefore, unlike data from prospective studies, the routinely available data are often clinical indication driven, and missing data are potentially not missing at random. For example, at entry, albuminuria status was not measured for over half of the cohort, and in the early years of data, the urine dipstick tests were widely used, with potential impact on reliability of albuminuria measures [32,33]. In our cohort, having unmeasured albuminuria was associated with higher cardiovascular risk compared to having it measured as normal (Tables 1 and 2), indicating that the category of patients with unmeasured albuminuria at entry is likely a mixed group of patients with different degrees of albuminuria. Similarly, we assumed that the absence of a record for a comorbid condition means the absence of the condition. This might not be the case with the absence of a diagnosis of diabetes or CVD, for example, potentially partially due to limitations in data recording or diagnosis

coding [34,35]. Furthermore, the vast majority of participants in the study cohort were in eGFR category G2, i.e., had only minor kidney function impairment and no CKD. This category of patients might be biased towards patients who had creatinine tests because of another condition (e.g., diabetes or hypertension) and, while covariates in the model will capture large differences in risk, model generalisability could be affected by omitted relevant factors. The much smaller number of participants with more advanced renal impairment, respectively, has likely contributed to more limited model discrimination in these categories. Further assessments of model performance in patient cohorts external to CPRD, including prospective clinical trials and observational cohorts with regular measurements of kidney function, could mitigate against the routine data limitations.

Our study has indicated areas where further development will be informative. Firstly, the evolving model complexity and related computational needs motivated us to limit the number of endpoints and relationships between them. Future model developments could explore utilising different model structures such as discrete event simulations. These are likely to facilitate the inclusion of larger number of model endpoints but may substantially increase model computation time and make uncertainty estimation less tractable [36]. Secondly, change in albuminuria over time is an important further marker of kidney function, and its inclusion in the model could enhance model performance and functionality [37]. Unfortunately, the limited albuminuria data in our database prevented us from following such an investigation.

In conclusion, the lifetime CKD–CVD policy model allows the simulation of long-term kidney function decline, cardiovascular morbidity, vascular and nonvascular mortality, health-related quality of life, and healthcare costs across the range of patients with reduced kidney function and overcomes many limitations of previous models. It will contribute to a greater understanding of the progression of kidney disease and its cardiovascular complications. In this paper, we have demonstrated its use to project disease risks and survival under particular treatment strategies, but its applications are wider. The model may be beneficial to health data scientists, health economists, and policy makers for comparative effectiveness and cost-effectiveness assessments in the evaluation of established and novel strategies for the management of patients with reduced kidney function.

## Supporting information

**S1 Checklist.**
(DOCX)

**S1 Text. Supporting information.**
(DOCX)

## Acknowledgments

We thank Dr Chris Jackson of the Medical Research Council (MRC) Biostatistics Unit, University of Cambridge for his advice on how to adapt functions from the msm package to our setting [38].

## Author Contributions

**Conceptualization:** Iryna Schlackow, Jason Oke, Richard J. Stevens, Rafael Perera, Borislava Mihaylova.

**Data curation:** Iryna Schlackow, Claire Simons, Christopher A. O'Callaghan, Daniel Lasserson.

**Formal analysis:** Iryna Schlackow, Claire Simons, Jason Oke.

**Funding acquisition:** F. D Richard Hobbs, Richard J. Stevens, Rafael Perera.

**Investigation:** Iryna Schlackow, Claire Simons, Borislava Mihaylova.

**Methodology:** Iryna Schlackow, Claire Simons, Jason Oke, Benjamin Feakins, Richard J. Stevens, Rafael Perera, Borislava Mihaylova.

**Project administration:** Borislava Mihaylova.

**Resources:** Christopher A. O'Callaghan, F. D Richard Hobbs, Daniel Lasserson, Borislava Mihaylova.

**Software:** Iryna Schlackow, Claire Simons, Jason Oke, Benjamin Feakins.

**Supervision:** Iryna Schlackow, Daniel Lasserson, Borislava Mihaylova.

**Validation:** Iryna Schlackow, Claire Simons, Christopher A. O'Callaghan, F. D Richard Hobbs, Daniel Lasserson, Richard J. Stevens, Rafael Perera, Borislava Mihaylova.

**Visualization:** Iryna Schlackow, Claire Simons, Benjamin Feakins.

**Writing – original draft:** Iryna Schlackow, Claire Simons, Borislava Mihaylova.

**Writing – review & editing:** Iryna Schlackow, Claire Simons, Jason Oke, Benjamin Feakins, Christopher A. O'Callaghan, F. D Richard Hobbs, Daniel Lasserson, Richard J. Stevens, Rafael Perera, Borislava Mihaylova.

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
