## [Decision Letter · Decision Letter 0]

9 Jul 2020

Dear Dr. Mihaylova,

Thank you very much for submitting your manuscript "Policy model for people with reduced kidney function and value of cardiovascular prevention" (PMEDICINE-D-19-04239) for consideration at PLOS Medicine. 

[LINK]

In light of these reviews, I am afraid that we will not be able to accept the manuscript for publication in the journal in its current form, but we would like to consider a revised version that addresses the reviewers' and editors' comments. Obviously we cannot make any decision about publication until we have seen the revised manuscript and your response, and we plan to seek re-review by one or more of the reviewers. 

We expect to receive your revised manuscript by Jul 30 2020 11:59PM. Please email us (plosmedicine@plos.org) if you have any questions or concerns.

We look forward to receiving your revised manuscript. 

Sincerely,

Adya Misra, PhD

Senior Editor 

PLOS Medicine

plosmedicine.org

Please revise your title according to PLOS Medicine's style. Your title must be nondeclarative and not a question. It should begin with main concept if possible. "Effect of" should be used only if causality can be inferred, i.e., for an RCT. Please place the study design ("A randomized controlled trial," "A retrospective study," "A modelling study," etc.) in the subtitle (ie, after a colon).

Abstract

Abstract Background: Provide the context of why the study is important. The final sentence should clearly state the study question.

Please include the study design, population and setting, number of participants, years during which the study took place, length of follow up, and main outcome measures. 

Please quantify the main results (with 95% CIs and p values). 

Please include the important dependent variables that are adjusted for in the analyses.

In the last sentence of the Abstract Methods and Findings section, please describe the main limitation(s) of the study's methodology.

Abstract Conclusions:

* Please address the study implications without overreaching what can be concluded from the data; the phrase "In this study, we observed ..." may be useful.

* Please interpret the study based on the results presented in the abstract, emphasizing what is new without overstating your conclusions.

* Please avoid vague statements such as "these results have major implications for policy/clinical care". Mention only specific implications substantiated by the results.

* Please avoid assertions of primacy ("We report for the first time....")

Author summary

References- please use square brackets and update the bibliography to Vancouver style. Please place the full stop after the square brackets. 

Lines 188-191 could you provide the sources of these external data please? It would be clearer to mention the names of data sets and provide citations.

Please can you ensure all acronyms are introduced on first view

Discussion

Please present and organize the Discussion as follows: a short, clear summary of the article's findings; what the study adds to existing research and where and why the results may differ from previous research; strengths and limitations of the study; implications and next steps for research, clinical practice, and/or public policy; one-paragraph conclusion.

Please include a brief summary of clinical implications of the model and your findings, as this section focusses heavily on the model development/validation aspects. 

Please ensure that the study is reported according to the TRIPOD guideline, and include the completed checklist as Supporting Information. When completing the checklist, please use section and paragraph numbers, rather than page numbers. Please add the following statement, or similar, to the Methods: "This study is reported as per the XXX guideline (S1 Checklist)."

Did your study have a prospective protocol or analysis plan? Please state this (either way) early in the Methods section.

Comments from the reviewers:

Reviewer #1: The authors describe a policy model they have developed for evaluating the long-term effects of cardiovascular interventions in patients with varying levels of reduced kidney function. This is very interesting work, which addresses highly clinically relevant questions. I was not able to fully follow how the models were developed based on the primary text and the supporting information. (I was not able to figure out how to access the Supplementary Information, which sounds like it might be been a separate document from the Supporting Information, however.) In part, this is likely because the modeling is complex, and it may not be possible to provide a complete description of the models and their assumptions within the scope of an article such as this. However, it is possible that some of the main concepts and assumptions underlying the models could be articulated more clearly, and that the authors may be able to better describe the impact of underlying assumptions of the models on their results. With this perspective, I'll list a few of the questions that occurred to me as I read the article in case they are of use to the authors:

1) It appeared from the description of the CKD submodel and the Integrated CKD-CVD model that the authors used the data from their cohort to estimate 1-year transition probabilities between the different CKD and CVD states as a function of various baseline covariates, and then used Markovian assumptions to model the state probabilities over time. However, the description of the CVD/non CV death submodel appears to be a Anderson-Gill survival model, was gives the impression of a model that relates the overall time-to-events to baseline covariates, as opposed to estimation of 1-year transition probabilities. This apparent contradiction in approaches left me confused.I think it would be useful to indicate if in fact all modeling for both the CKD and the CVD states was based on development of 1-year transition probabilities between states, where the transition rates were allowed to depend on the covariates that were indicated in the manuscript. And if so, how did the Anderson-Gill analyses for the CVD/non CV death model inform this transition rates?

2) How were the different types of events accounted for in the Anderson Gill models that evaluated multiple event times?

3) If in fact the modeling was done based on 1-year transition probabilities between CVD and CKD states, I would have thought that it would also be necessary to model longitudinal changes from year to year in the various patient characteristics that inform the transition probabilities. Alternatively, if the transition rates are model in terms of the baseline patient characteristics only, I would think it would be necessary to allow the transition rates (for given values of the baseline characteristics) to change over time. For example, sure the risk of transitioning to a worse state at a given baseline proteinuria would differ for the 1st year and the 20th year after baseline. But I didn't see any of this mentioned in the text. I think it would be useful to clarify.

4) It appears the Markov models were based on transition probabilities assuming none of the CVD interventions were in place for the untreated condition, and then the authors incorporate estimates of the treatment effects for these interventions based on the literature to estimate transition probabilities for the treated patients. Since many of the patients in the actual cohort were treated, how were transition probabilities for untreated patients estimated?

5) When incorporating treatment effects from the literature, were treatment effects assumed constant for all patients on a hazard ratio or relative risk scale, as is commonly assumed? Or were they assumed constant on an absolute risk scale? Or was some other assumption used? Were any treatment by covariate interactions assumed, or were the treatment effects assumed constant (on one of the scales noted above) across different patient characteristics? Were the treatment effects assumed constant over time, or was some attenuation in the effect of the treatment over time assumed? These all seem like very fundamental assumptions that are likely to strongly affect the results that were presented, and I think it may be useful to summarize the assumptions that were made, and their likely implications for the results that were presented.

Reviewer #2: Thanks for the opportunity to review your manuscript. My role is as a statistical reviewer so my queries and comments are concentrated on the analysis and data, and the reporting of these. This is a very interesting manuscript, which selects of a cohort of people with impaired kidney function and then uses the outcomes of this cohort to predict CKD and CV and uses these risk prediction models in a Markov model to look at implications different treatment options. The manuscript is clearly written and with useful supplementary information about the models underlying the key results reported. I have some overall queries about the analysis, followed by ones specific to a section of the manuscript.

An overall query I had was the whether the intention of the manuscript methodological (development and validation of the tool) or is it concentrated on the results of the Markov model? I appreciate that this is a complex analysis and with limited space it's difficult to give a detailed context for kidney impairment and the current state of treatment options for patients. There is probably a series of papers that could come from this work (a 'treatment gap' analysis would be very interesting). 

I may have missed it but I couldn't find a table describing the patterns of missingness for the covariates used in the study by eGFR status - Table SM1 is helpful and if possible to get this by the kidney function patients this gives a sense of limitations of the CPRD data. The approach to missing data is explained well, and the limitations of the possibility of data being MNAR is also acknowledged in the discussion. I don't think there is an obvious MNAR sensitivity analysis that could be implemented with this data and analysis. 

Only 5 imputations were used, this is a relatively low number. What criteria was used to decide the number of imputations, i.e. a quantitative measure like fraction of missing information or was the small number of imputations necessary because of the size of the dataset (my own personal experience is that both Stata and R are very capable with MI but the process is extremely time-consuming with a large dataset). 

Are the risk prediction models available for use by other researchers, e.g. as PMML or as R-code? Similarly is the code for the other models (CKD submodel and Markov model) available?

The c-index shows that the predictive models perform better in patients with better kidney function. What are the consequences for the Markov models when the discrimination of the risk models is lower? 

P5, L100. Is the 'up to standard' a general quality measure for CPRD data or is this specific to this study?

P5, L106. Is natural decline in kidney function with age incorporated into the selection criteria or is this likely to have a minimal impact on finding cases?

In Supp Fig 1 there are ~75k patients excluded as they had abnormal tests less than 90 days apart - is this used as an indicator of AKI or suspected AKI? When results are this close together are the patients then excluded completely (even if they have a long-term pattern of kidney impairment) or are these tests just not considered to be evidence of CKD? My understanding is that long-term loss of function is considered to be a potential outcome of AKI even after recovery in the short-term.

P5, L115. Could a patient be censored when they transfer out of one practice, and then enter the cohort (as a 'different' patient) again if they meet the inclusion criteria at a new practice?

P5. L135. Does this refer to missing uACR being accounted for using the missing-as-indicator approach?

P5, L152. What specific algorithm using AIC was used? 

P5. L153. Was an alpha of 0.01 used as a cut-point to determine statistical significance? This is mentioned a few lines later but I couldn't tell if this was the criteria used in this step.

What was the criteria for 'clinical' significance'?

P5. L157. So model selection was done with the recurrent-events analysis, and then a parametric survival model was estimated including the variables identified from the initial model selection?

P10, L213. Typo in 'szmpled' - should be sampled?

Reviewer #3: See attached file.

[LINK]

---

## [Decision Letter · Decision Letter 1]

2 Nov 2020

Dear Dr. Mihaylova,

Thank you very much for re-submitting your manuscript "Long-term health outcomes of people with reduced kidney function: a policy model derived from a large UK population cohort" (PMEDICINE-D-19-04239R1) for review by PLOS Medicine.

I have discussed the paper with my colleagues and the academic editor and it was also seen again by xxx reviewers. I am pleased to say that provided the remaining editorial and production issues are dealt with we are planning to accept the paper for publication in the journal.

[LINK]

We look forward to receiving the revised manuscript by Nov 09 2020 11:59PM. 

Sincerely,

Adya Misra, PhD

Senior Editor 

PLOS Medicine

plosmedicine.org

Requests from Editors:

Title : please revise to "Long-term health outcomes of people with reduced kidney function in the UK: a modelling study" to include a study descriptor in accordance with PLOS Medicine style

Abstract-Please provide an age range for participants included in your cohort

Abstract- limitations should be explicitly noted at the end of the methods and findings, for example “limitations include xxx”. Please also provide 2-3 limitations

Abstract- could you define “optimal cardioprotective medication use”?

"Substantial benefit" is perhaps a bit of an overstatement that should be toned down in the abstract and elsewhere

towards the end of the abstract, suggest that substituting actual numbers for "more than doubled" and "increased by a third" would be better

Author summary

Line 5-6 could you revise to specify for whom this is a key target?

Line 15-16 needs to be revised for clarity, please specify what categories were used and what model performance means?

Page 9 line 12- I suggest this is either removed or moved to the data availability statement. PLOS data policy requires that the data underlying the findings be provided in a repository or as supplementary information unless there are ethical restrictions to data-sharing. If there are restrictions on sharing codelists and algorithms, please provide these in the data availability statement also noting the contact details for who may be able to provide these to interested parties. Please note the sole contact cannot be one of the study authors.

Page 20 line 4-6 please rephrase “likely to be used by” to “may be beneficial to” to avoid overstating utility

Please add an additional sentence to the paragraph beginning the discussion to summarize the study's findings

Funding information/competing interests and data availability statements can be removed from the main text as these are provided in the article meta-data

Please provide access details for ref 32

Comments from Reviewers:

Reviewer #1: The authors have nicely addressed the comments of the initial review. 

Reviewer #2: Thanks for the revised manuscript and comprehensive replies to my original queries.

My original queries have been answered or dealt with by changes to the manuscript. The availability of the code should be useful as I think there will be interest in applying a similar model to CKD/reduced kidney function patients in other countries. 

I appreciate that with a very large dataset there are practical limitations to applying MI in dealing with missing data and the approach used is a reasonable compromise (dealing with CPRD data with many imputations is likely to need more memory than most computers have). 

For a dataset of this size the use of p<0.05 for variable selection and p<0.01 for 'significance' is a reasonable choice as well. 

Reviewer #3: The authors have done an excellent and thorough job at addressing all reviewer comments. I have no further concerns.

[LINK]

---

## [Editor Report · Decision Letter 2]

30 Nov 2020

Dear Dr. Mihaylova, 

On behalf of my colleagues and the academic editor, Dr. Maarten Taal, I am delighted to inform you that your manuscript entitled "Long-term health outcomes of people with reduced kidney function in the UK: a modelling study using population health data" (PMEDICINE-D-19-04239R2) has been accepted for publication in PLOS Medicine. 

PRODUCTION PROCESS

Before publication you will see the copyedited word document (within 5 business days) and a PDF proof shortly after that. The copyeditor will be in touch shortly before sending you the copyedited Word document. We will make some revisions at copyediting stage to conform to our general style, and for clarification. When you receive this version you should check and revise it very carefully, including figures, tables, references, and supporting information, because corrections at the next stage (proofs) will be strictly limited to (1) errors in author names or affiliations, (2) errors of scientific fact that would cause misunderstandings to readers, and (3) printer's (introduced) errors. Please return the copyedited file within 2 business days in order to ensure timely delivery of the PDF proof. 

If you are likely to be away when either this document or the proof is sent, please ensure we have contact information of a second person, as we will need you to respond quickly at each point. Given the disruptions resulting from the ongoing COVID-19 pandemic, there may be delays in the production process. We apologise in advance for any inconvenience caused and will do our best to minimize impact as far as possible.

EARLY VERSION

PRESS

PROFILE INFORMATION

Thank you again for submitting the manuscript to PLOS Medicine. We look forward to publishing it. 

Best wishes, 

Adya Misra, PhD

Senior Editor 

PLOS Medicine

plosmedicine.org